# Anatomy and evolution of the first Coleoidea in the Carboniferous

Christian Klug [1], Neil H. Landman[2], Dirk Fuchs[3], Royal H. Mapes[2], Alexander Pohle [1], Pierre Guériau [4,5], Solenn Reguer[5] & René Hoffmann[6]

Coleoidea (squids and octopuses) comprise all crown group cephalopods except the Nautilida. Coleoids are characterized by internal shell (endocochleate), ink sac and arm hooks, while nautilids lack an ink sac, arm hooks, suckers, and have an external conch (ectocochleate). Differentiating between straight conical conchs (orthocones) of Palaeozoic Coleoidea and other ectocochleates is only possible when rostrum (shell covering the chambered phragmocone) and body chamber are preserved. Here, we provide information on how this internalization might have evolved. We re-examined one of the oldest coleoids, *Gordoniconus beargulchensis* from the Early Carboniferous of the Bear Gulch Fossil-Lagerstätte (Montana) by synchrotron, various lights and Reflectance Transformation Imaging (RTI). This revealed previously unappreciated anatomical details, on which we base evolutionary scenarios of how the internalization and other evolutionary steps in early coleoid evolution proceeded. We suggest that conch internalization happened rather suddenly including early growth stages while the ink sac evolved slightly later.

[1] Paläontologisches Institut und Museum, Universität Zürich, Karl-Schmid-Strasse 4, 8006 Zürich, Switzerland. [2] Division of Paleontology (Invertebrates), American Museum of Natural History, 79th Street and Central Park West, New York, NY 10024, USA. [3] SNSB-Bayerische Staatssammlung für Paläontologie und Geologie, Richard-Wagner-Straße. 10, 80333 Munich, Germany. [4] IPANEMA, CNRS, Ministère de la Culture, UVSQ, USR3461, Université Paris–Saclay, 91192 Gif-sur-Yvette, France. [5] Synchrotron SOLEIL, 91192 Gif-sur-Yvette, France. [6] Institute of Geology, Mineralogy, & Geophysics, Ruhr-Universität Bochum, 44801 Bochum, Germany. Correspondence and requests for materials should be addressed to C.K. (email: chklug@pim.uzh.ch)

Temporally close to the origin of a clade, the number of species and specimens belonging to that clade is often low. In soft-bodied organisms with a low preservation potential, the subsequent number of fossil specimens is often less than a handful hampering the recognition of such early representatives since characters are usually not as differentiated as in more derived species. Hence, the origin of the Coleoidea is plagued with a number of unresolved questions.

The currently most widely accepted phylogenetic hypothesis for the origin of coleoids suggests the extinct Bactritoidea as sister group[1]. In turn, the bactritoids share the straight conical conch with their orthocerid ancestors, but they differ in the position of the siphuncle connecting the phragmocone chambers; in orthocerids, the siphuncle is more or less in a central position, while in bactritoids and many early coleoids, it is ventral. This character state is maintained in early coleoids, including all Paleozoic forms and all Belemnitida, while it becomes altered or reduced (with the reduction of the phragmocone) in octopuses and most squids.

Some cephalopod remains from the Early Devonian were interpreted as coleoids based on the similarity of their body chambers to those of two species of much younger phragmoteuthids (Triassic) and based on the presence of a rostrum in another species[1]. In spite of similarities to the few unambiguous Carboniferous coleoids, some authors have questioned the coleoid nature of these Devonian forms[2–7]. While the absence of coleoid fossils from the Middle and Late Devonian casts doubt on the coleoid nature of these Early Devonian remains, orthoconic cephalopod conchs are poorly studied and thus, early coleoids may have been frequently overlooked or misinterpreted as bactritoids[8]. Several unequivocal coleoid fossils are known from the Carboniferous and some are even exceptionally well-preserved, thereby providing a great amount of anatomical information[5,7,9–16]. Most studies on molecular clock data place the divergence of the major coleoid clades (and thus the origin of the crown group) in the Permian or Triassic[7,17–21]. However, taking the large confidence intervals in these studies into account, even a younger or older divergence date is conceivable and the timing of the origin of coleoids has presently not been accurately resolved by molecular studies.

In this context, the cephalopod remains from the Bear Gulch Limestone Konservat-Lagerstätte (Late Mississippian: Late Chesterian = Late Serpukhovian[22]) of Bear Gulch in Montana[23–27] are essential for our understanding of coleoid evolution. Here, we focus on the type material of *Gordoniconus beargulchensis*[28]. It preserves the complete orthoconic conch with its conical phragmocone, a long body chamber with a slightly constricted and curved aperture, and a conical rostrum. Like all the cephalopods from this conservation deposit, none of the original aragonitic shell and rostrum retains the original mineralogy and the holotype was first described in open nomenclature[9]. The flattened specimen was initially described as an orthocerid with the description focusing on the stomach contents (conodonts, fish remains) and the jaw. However, with the presence of both an ink sac and a rostrum, Mapes et al.[5,28] recognized the coleoid nature of the specimen and added some morphological details based on standard microscope examinations. When it came to the soft part imprints around the jaw, they limited their interpretation to the words "*cephalic region*"[29].

We re-examined the specimen using lights of varying wavelengths, including ultraviolet (UV)-light and white light applied at a very shallow angle, including Reflectance Transformation Imaging (RTI), as well as synchrotron micro-X-ray fluorescence major-to-trace elemental mapping. The imagery allowed several previously unappreciated anatomical details to be observed. Here, we describe these details, discuss them in the evolutionary context of related cephalopod groups and reflect on possible processes

that led to the internalization of the conch and thus coleoid origin. This enabled us to reconstruct the evolutionary innovations involved with the origination of the Coleoidea such as conch internalization followed by the evolution of the ink sac and possibly further differentiation of the brain.

## Results

**Material.** The main specimen consists of four fragments of one individual, the holotype of *Gordoniconus beargulchensis*. Both part and counterpart are broken in the middle (formerly glued). The specimen is kept with the numbers AMNH 43264 (part) and AMNH 50267 (counterpart) in the American Museum of Natural History in New York (Fig. 1). We refer to the parts with the body chamber and soft parts as the anterior part and the part with phragmocone and rostrum is dubbed posterior part. We consider the parts, where the fossil forms a convex structure (elevated above the bedding plane), as the part (AMNH 43264) and the concave side as the counterpart (AMNH 50267). In addition, we examined the paratype (AMNH 5743) of *G. beargulchensis* as well

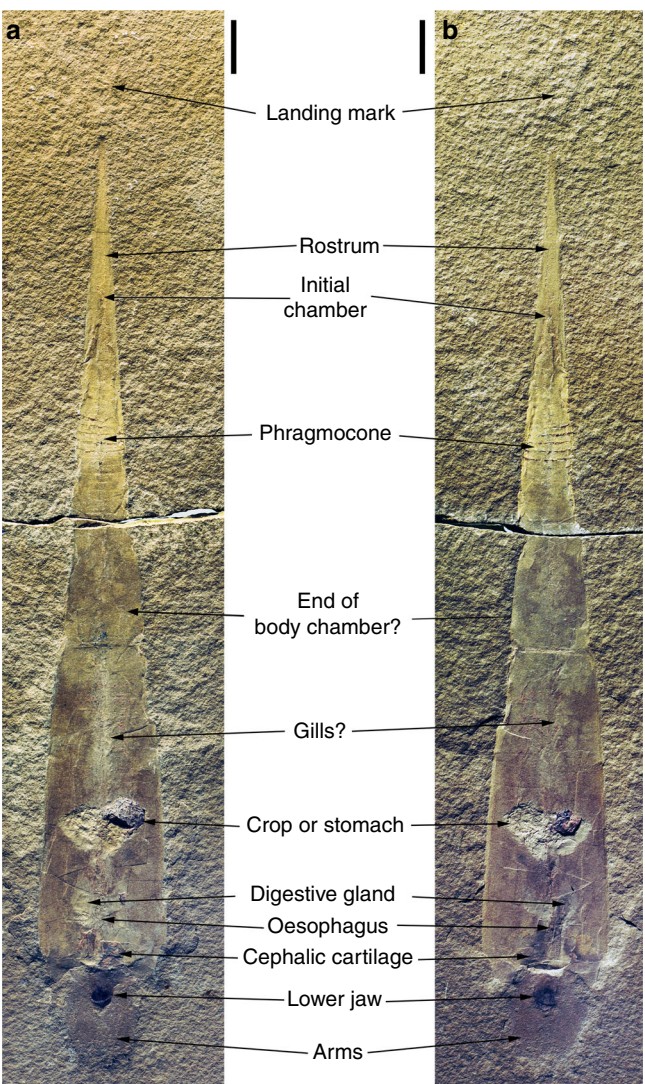

**Fig. 1** Holotype of *Gordoniconus beargulchensis*. Part and counterpart of the best preserved specimen of one of the oldest known coleoid fossils, displaying the shape of its complete hard parts and remains of the most important non-mineralized organs. Scale bar is 10 mm. **a** Counterpart (AMNH 50267) and **b** part (AMNH 43264), Bear Gulch Limestone (Late Mississippian), Bear Gulch, Montana (USA)

as some topotypes for comparison from the Carnegie Museum of Natural History, Pittsburgh, Pennsylvania (CM) and the University of Montana Paleontology Center Research Collection, Missoula, Montana (UMPC).

**Arms**. In front of the aperture, a structure with irregular outline measuring 15 x 15 mm under UV-light (brownish; Fig. 2f–j), in the RTI-images (Supplementary RTI file) and 18 mm (length) x 15 mm (width) under white light is visible in anterior part and counterpart (AMNH 43264, 50267; Fig. 2a–e). Under white and UV-light, a 6.5 mm long and 4.2 mm wide dark oval structure is visible, which was earlier interpreted as the jaws[5,9,28]. The surrounding two lobes were called cephalic region by Mapes et al.[4] and they wrote that tentacle stumps do not cause the irregularity of its outline.

We re-examined the cephalic region using RTI (Supplementary RTI file) and white light applied at a very low angle to enhance contrast (Fig. 2a–e). The photos revealed longitudinally oriented furrows subdividing this field into 5 to 7 longitudinal structures. These structures converge slightly towards the plain of symmetry anterior of the mouthparts. At the anterior end, this causes an irregular outline with about four to six subtriangular elevations in the part and furrows in the counterpart of around 2 mm length.

Are these structures arm crown-remains? First, the position in relation to other organs fits. The supposed arms lie outside the aperture and anterior to the buccal mass. Additionally, they slightly converge anteriorly; when examining the venter of soft parts of Recent nautilids, the arms do the same around the hyponome[29,30]; similarly, the arms converge around the funnel in modern *Spirula*[31,32]. Second, the low number of arms coincides with the supposed plesiomorphic state in coleoids. The earliest other coleoids from the Carboniferous with preserved arms (*Jeletzkya douglassae*[33] and questionably *Pohlsepia mazonensis*[34]) are from the Pennsylvanian and have ten arms as do Mesozoic stem group coleoids (belemnites, diplobelids and phragmoteuthids[13,33,35–38]). Soft part preservation in orthoconic cephalopods is exceedingly rare. However, the few that are known also fit the supposed plesiomorphic state of a low number of arms[39]. Third, the proportions of the arms in relation to the buccal mass, conch length and width are within the broad range of proportions in living coleoids. In *Magnapinna*, the arms can be ten times as long as the mantle, while in *Teuthowenia*, the mantle is five times longer than the arms. Contracted arms of *Spirula* measure a sixth of its mantle length and in Jurassic *Plesioteuthis*, contracted arms measure roughly 10% of mantle length[40]. In *Gordoniconus*, the arms measure about 10% of conch and mantle length. Additionally, the proportion of arm length to width is within the range of Recent coleoids. Taking these facts together, we conclude that all homology criteria are fulfilled and thus, arms are preserved in the holotype of *Gordoniconus*.

Because of the absence of arm hooks and the faintness of the separating furrows between the arms, it is difficult to determine the number of arms. Presuming that the arm crown was arranged circumorally as in Recent coleoids, we assume that not all arms are visible in ventral or dorsal aspect. Thus, at least half of the arms should be visible from below. Both in slab and counter slab, about six arm imprints of similar length can be seen. It appears unlikely that only two arms are covered by these six arms and thus, we infer that ten short arms were present. In the soft parts of the holotype, we do not see evidence for modified tentacles (or their bases), which should be longer than the arms, thus conforming to the supposed plesiomorphic state.

Mapes et al.[28] assigned a supposedly hook-bearing tentacle from a different specimen (paratype UMPC 5743) to this species. We doubt that it belongs to the same species, because the

associated phragmocone has a higher apical angle and an ink sac, absent in all other specimens of *Gordoniconus* we examined. The evidence presented there is not convincing, because the alleged arm hooks are irregularly shaped and arranged. A re-examination (Fig. 3) revealed that this structure is a coprolite containing irregular fibers that superficially resemble arm hooks. If the phylogeny of Kröger et al.[1] is accepted, tentacles (modified arm pair IV) probably evolved after the Paleozoic. Also, at least the bases of the tentacles should be preserved because, as demonstrated by Clements et al.[41], if soft parts are fossilized in coleoids, this usually covers particularly arms and tentacles. Accordingly, we suggest that *Gordoniconus* had ten arms of similar proportions.

**Funnel (hyponome)**. The anterior convergence of the arms is likely caused by the funnel. Since all crown group cephalopods have a central ventral funnel, this was likely also the case in *Gordoniconus*. There is one structure preserved posterior of the buccal mass that might either represent remains of the funnel or of the cephalic cartilage (Fig. 2k–o).

**Buccal mass**. Preservation, position, and proportions corroborate the interpretation of the dark oval structure anterior of the aperture as jaw remains (Figs. 1 and 2). All previous authors came to the same conclusion[5,9,28]. They noted that the anterior part of the jaw is black, i.e. darker than the posterior part. Additionally, there is an anterior crescent-shaped field (Fig. 2) that is morphologically separated from the lighter colored posterior part. As we support that the arms converge anterior to the supposed funnel (hyponome), then the specimen is seen in ventral view with the larger lower jaw being visible. Thus, the crescent-shaped structure can be interpreted as the short outer lamella (characteristic for lower jaws) with the lighter colored and much longer inner lamella behind. In the counterpart, the inner lamella shows a lighter colored central stripe (Fig. 2k–o). This is either a taphonomic artifact or an imprint of the upper jaw.

**Cephalic cartilage**. Examination of the material under UV-light (Fig. 2i, j) and light of different visible wavelengths (Fig. 2k–o) revealed several structures invisible otherwise (Fig. 2). Under UV-light, a series of structures appears in orange (Fig. 2a–e). From its color under white and UV-light, we infer that these structures represent phosphatized soft-tissue-remains. Phosphatisation of these structures is corroborated by their incorporation of strontium and most importantly yttrium that preferentially substitute for Ca in calcium phosphate[42] as shown in the µXRF element maps (Fig. 4).

The largest of these structures is the crop or stomach content, which contains prey remains[9]. There are numerous small orange patches in the UV-images posterior and anterior of the crop/stomach (Fig. 2f–j). Anteriorly, some of these patches are aligned along the midline, suggesting affiliation with the esophagus. Directly posterior to the aperture, there is an irregular, asymmetrical, bilobate structure, each of the lobes measuring 4 to 6 mm in length and 4 to 5 mm in width. Since the two lobes have roughly the same size, outline and appearance under UV-light as well as, e.g., reflectance under visible and infrared lights (Fig. 5a, e), the asymmetry might be due to taphonomic alteration and the structure might originally have been symmetrical. Since these two lobes lie immediately posterior of the aperture and jaws, we assume that they still belong to the head region and possibly represent remains of the cephalic cartilage[37,43–45]. Alternatively, they could represent parts of the digestive glands, but this hypothesis appears less likely: due to its chitinous cover, the cephalic cartilage has a preservation

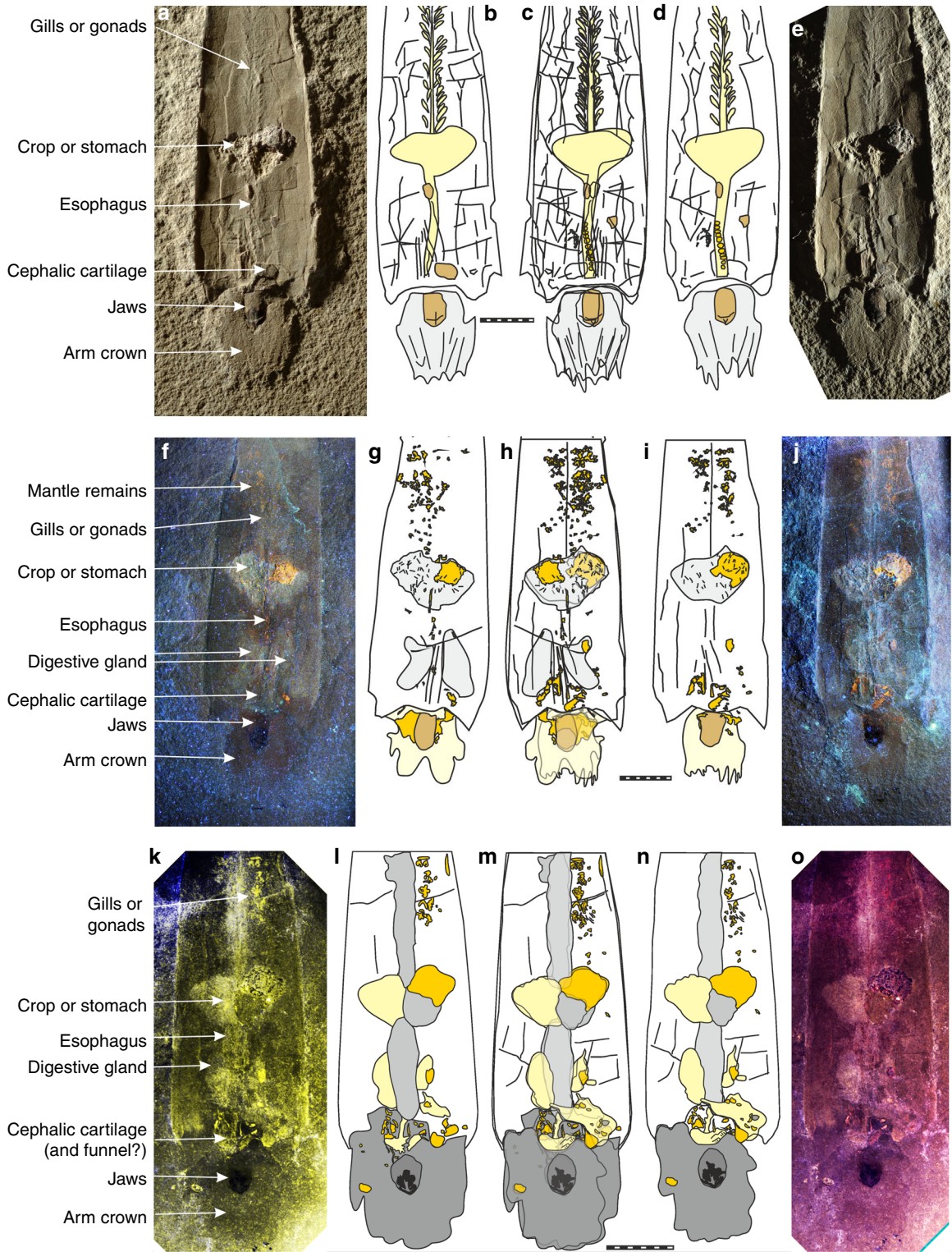

**Fig. 2** *Gordoniconus beargulchensis*, counterpart (AMNH 50267, **a–c**, **f–h**, **k–m**) and part (AMNH 43264, **c–e**, **h–j**, **m–o**), photographed using lights of different wavelengths to make fossilized soft parts visible. Scale bar is 10 mm. **a–e** Photographed under white light at an acute angle. **b–d** Drawings of the visible structures. **c** Combination of **b** and **d**. **f–j** Photographed under UV-light. **g–i** Drawings of the visible structures. **h** Combination of **g** and **i**. **k–o** False-color overlays of reflectance under blue (435 nm) and infrared (770 nm) lights (**k–m**) and under blue (435 nm), green (550 nm), and red (660 nm) lights (**m–o**). **l–m** Drawings of the visible structures. **m** combination of **l** and **n**

potential almost as high as that of the jaws and the esophagus and thus has a higher preservation potential than the digestive organs due to recalcitrance of the different tissue histologies. The size of these two structures contradicts an interpretation as statocysts in modern cephalopods, but it is conceivable that statocysts are a part of it.

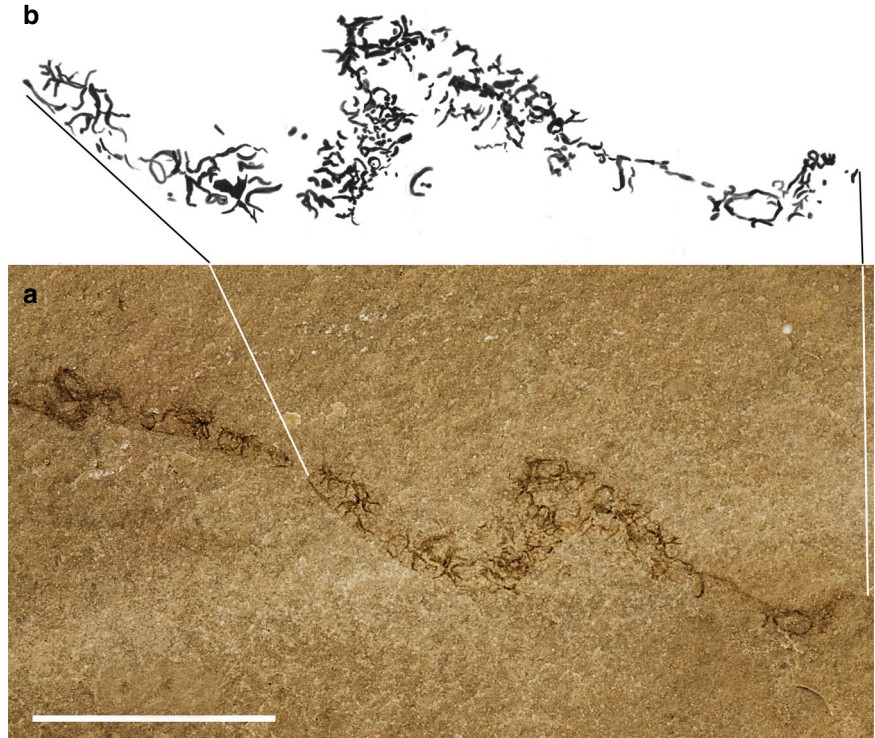

**Fig. 3** Part of a structure previously interpreted as a tentacle of *Gordoniconus beargulchensis* (paratype, UMPC 5743) photographed under white light (**a**) and parts of it drawn by Mariah Slovacek (**b**). Scale bar is 10 mm. Owing to the irregularity of the supposed arm hooks, its preservation and arrangement, we suggest that this is a trace fossil and not a tentacle

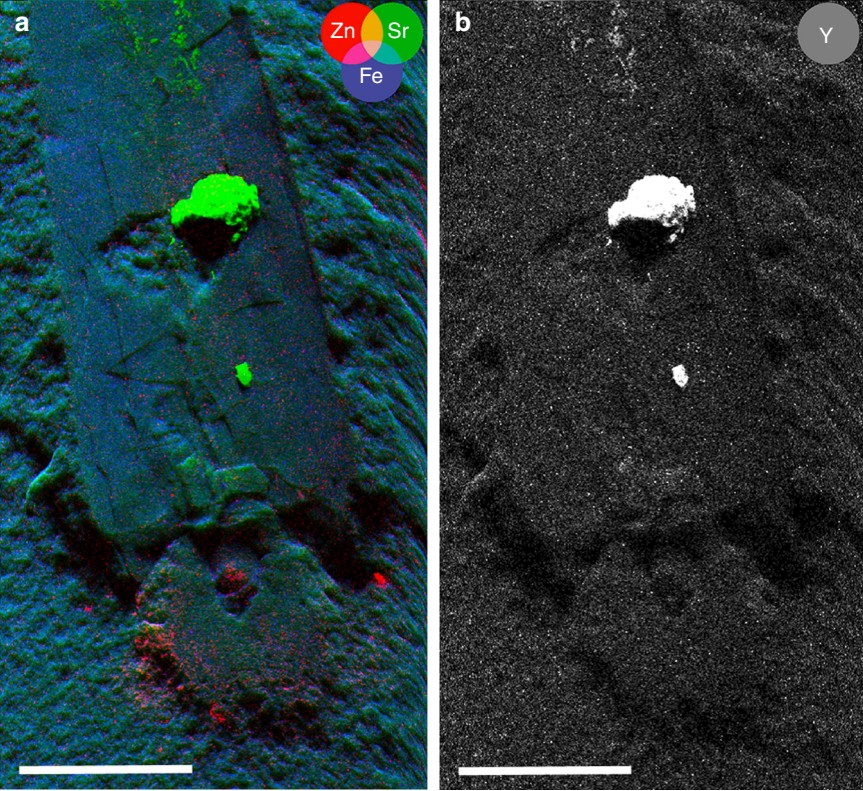

**Fig. 4** Synchrotron X-ray fluorescence mapping of the anterior part of the holotype of *Gordoniconus beargulchensis*, AMNH 50267. Scale bar is 10 mm. **a** False color overlay of zinc (red), strontium (green), and iron (blue) distributions reconstructed from Kα peaks integrated intensities. **b** Yttrium distribution. The color scale goes from dark (for low intensity) to bright (high intensity) going through pale. Strontium and yttrium highlight the phosphatized tissues (green and white spots), whereas the arm crown and jaws appear enriched in zinc. This is corroborated by observation under UV-light

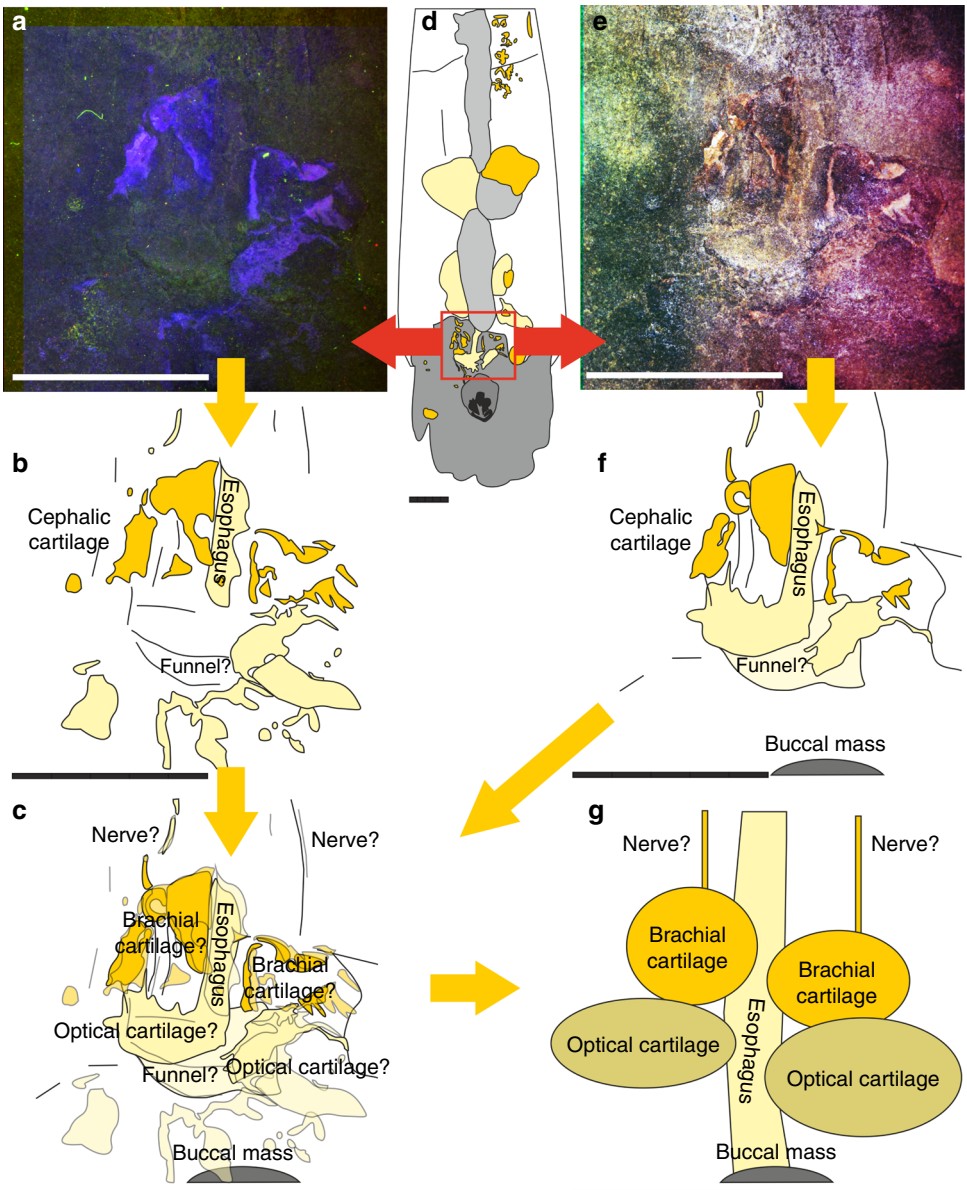

**Fig. 5** Morphological details of the cephalic cartilage of *Gordoniconus beargulchensis* (AMNH 50267). Scale bars are 5 mm long. The red and yellow arrows indicate the logical arrangement of the figures: red relates **a** and **e** to the rectangle in **d** and the yellow arrows show the sequence of interpretation and combination of imagery. **a** False-color overlay of luminescence images collected at 435, 571, and 935 nm under UV-A (365 nm) excitation. **b** Sketch of the structures seen in **a**. **c** combination of **b** and **f**. **d** Overview, red square shows the area of interest depicted in **a** to **c** and **e** to **g**. **e** False-color overlay of reflectance images collected under blue (435 nm), green (550 nm), and red (660 nm) excitations. **f** Sketch of the structures seen in **e**. **g** Tentative reconstruction of parts of the structures of the cephalic cartilage

The visible structures (Fig. 5a, e) appear like thin sheets of phosphate. Their superficial appearance suggests that the structures are subdivided in a roughly symmetrical anterior portion that reflects colored light less strongly and a pair of posterior structures that shows fine folds and reflects the light more strongly. All these parts surround the supposed esophagus. Its position around the esophagus directly posterior to the buccal mass corroborates their interpretation as cephalic cartilage. Additionally, this position within the body chamber implicates a retractable head.

**Digestive tract**. The crop or stomach was already recognized in the first description[9] with its contents of fish and conodont remains. It is 8 mm long, 15 mm wide and filled by an amorphous light gray phosphatic mass containing dark brown skeletal prey remains.

A straight structure that connects the crop with the jaws lies anterior to the crop; it is visible under white light (Fig. 2a–e) and shows a series of orange spots under UV-light (Fig. 2f–j). It is 20 mm long and about 1 to 2 mm wide. Under white light, this structure shows a series of small constrictions and elevations (Fig. 2a–e). Its proportions, dimensions, the slight phosphatisation and the connection with crop and jaw, as well as the fact that it crosses the supposed cephalic cartilage corroborate its interpretation as esophagus.

Another structure appears faintly in photos taken under UV- to IR-lights (Fig. 2g–o). It has an outline reminiscent of a butterfly and is about 12 mm long and each of the two sides about 5 mm wide. This could be an artifact, but since it is visible under light of various wavelengths, we think that it represents a fossilized organ. If true, it could be the digestive glands according to their relatively

large dimensions, their position attached to the esophagus behind the cephalic cartilage and their overall proportions (elongate, arranged parallel to the esophagus).

**Ink sac**. In the examined material of *Gordoniconus*, there is no evidence for the presence of an ink sac. By contrast, Mapes et al.[28] depicted a rather big ink sac in another specimen of probably a different species from the same locality and age (UM 5743). Its outline and anatomical position cannot be reconstructed because this specimen does not show a lot of further detail. The large size of the ink sac—missing in the better preserved holotype—casts doubt on the taxonomic assignment. Additionally, we think that the ink sac evolved with the gradual reduction of the conch, which is not visible in *Gordoniconus*. Accordingly, the absence of an ink sac might be taphonomic in the examined material or UM 5743 belongs to a different species, which is supported by the higher apical angle (20 instead of 10° in the flattened state) in the specimen with ink sac and the possible absence of a rostrum. We examined four well-preserved topotypes of *Gordoniconus* from the Carnegie Museum of Natural History (Pittsburgh), which also show soft-tissue preservation like the holotype but no ink sac (Supplementary Figs. 1–3).

**Gills or gonads?**. Behind the crop, there is a structure visible under white light (Fig. 2) and in the RTI-images (Supplementary RTI-file). It is approximately symmetrical, elongate and spans most of the distance between the crop and the last phragmocone chamber. It is 25 to 30 mm long and about 5 mm wide. There is a central ridge, almost 2 mm wide at its anterior end, to which about 20 lanceolate lappets 3 mm long and 1 mm wide are attached (Fig. 2a–e).

Superficially, this structure resembles the gills as preserved in Mesozoic coleoids[36,40]. The position casts doubt on this interpretation: in modern coleoids, the gills are usually situated in the mantle cavity ventral to the digestive tract and not posteriorly; in the posterior portion of the soft body, there is the stomach and the gonads behind it. Since gonads do not show a filamentous structure in modern coleoids, we tentatively interpret them as gills[40,46]. In the area where the gills or gonads are preserved, a series of spots is visible under UV-light and other wavelengths (Fig. 2f–o). These spots are irregular in size, shape and arrangement and thus, it is unclear whether these structures represent parts of the gills, gonads, mantle or something else. A posterior position of the gills appears disadvantageous as being distal to the mantle aperture; it would thereby decrease oxygen-rich water supply and increase metabolic waste products. Applying homology criteria to the alternative interpretations, the criterion of structure is fulfilled by gills; the criterion of position is fulfilled for gonads, the digestive tract and with some reservation for gills (somewhat similar in nautilids); the criteria of embryology and continuity cannot be applied due to lack of data. Thus, we prefer the interpretation as gills because their structure resembles other fossilized coleoid gills[40,46].

**Mantle**. Unlike in post-Triassic coleoid fossils[41,47], there are no thick striated remains of muscular mantle. This is not surprising because *Gordoniconus* represents a very early evolutionary stage of conch internalization, reduction of hard parts, as well as enhancement of the efficiency of the locomotory apparatus. Nevertheless, within the body chamber, many small orange patches can be seen under UV-to-IR lights, particularly behind the crop/ stomach (Fig. 2). They lack a defined structure, arrangement or anatomical detail, thus hampering their homologisation with organs of modern relatives. Nevertheless, they could represent skinny mantle remains (or dermis[41]), since it is an important

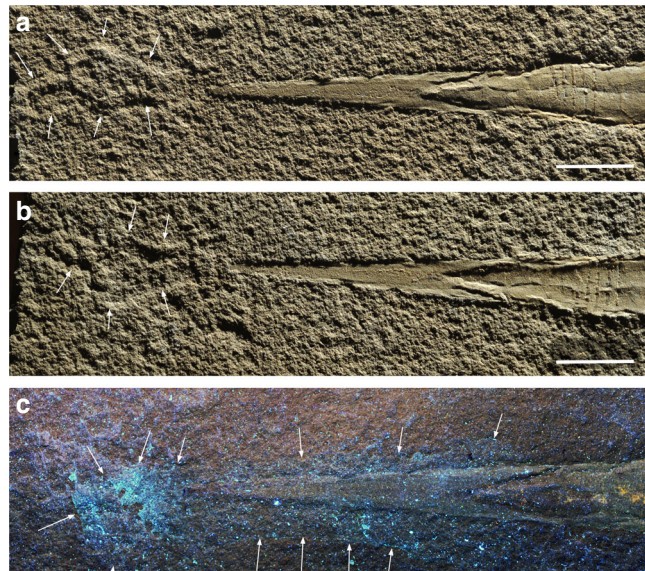

**Fig. 6** Photos of the posterior part of the holotype of *Gordoniconus beargulchensis*, counterpart (AMNH 50267, **a**, **c**) and part (AMNH 43264, **b**, **c**). The striped triangular (originally conical) part to the right is the chambered phragmocone, the arrow-shaped surface in the center of the image is the flattened rostrum, to which fins were attached in more derived coleoids. The imprints on the left in the photos were first thought to be fin remains. Scale bar is 10 mm. **a**, **b** Photographed under white light; arrows point at the limits of the landing mark. **c** Under UV-light; the white arrows point at the limits of the remains of glue or varnish that shine blue in UV-light

and large organ that has a great potential to become phosphatized as evidenced by Mesozoic relatives[36,41,42,47].

**Fins or landing mark?**. Although visible in white light, the blue patch that appears under UV-light (Fig. 6c) behind the rostrum is probably a film of glue or varnish and thus an artifact. Therefore, we base the discussion on the images taken under white light at a very low angle (Fig. 6a, b). The symmetrical structure lies 5 mm behind the posterior tip of the rostrum, it is 10 mm wide and almost 20 mm long. The anterior 12 mm form a pear-shaped imprint broadening posteriorly (Fig. 6). We suggest two alternative interpretations for this structure; either, these could be (1) the fins or (2) it is a landing mark that formed when the carcass hit the sediment with the tip of the rostrum. Such landing marks are well-known from Jurassic coleoids[37,40,48]. We do not have conclusive arguments to decide on this matter because anatomical detail or details classifying it as a landing mark are missing. Nevertheless, as none of the different imaging techniques revealed a particular composition consistent with a soft part nature, we think that it is a landing mark.

## Discussion

What was the sequence of evolutionary events? The preservation of the head-foot complex with the arm crown of short and morphologically similar arms without tentacles is an important finding; based on the arrangement, we suppose that *Gordoniconus* had probably ten arms. If accepting the decabrachian (ten-armed) condition as plesiomorphic for the Coleoidea, the Bactritida are likely to have shared this character state. Combining this with the phylogenetic bracket[40], all Orthocerida, Ammonoidea, and early Nautilida (depending on the phylogenetic hypothesis) should have shared this character state. In modern Nautilida, the ten-

armed condition is expressed only early in embryogenesis[30]. Possibly, early stem group nautilids also had ten arms as adults.

When did the differentiation into eight arms and two tentacles occur? A differentiation in arm dimensions and function might have occurred easily and thus also early. It could be related to a more active predatory mode of life in which the two tentacles aided in prey capture and manipulation. If correct, the appearance of tentacles may coincide with the development of a stronger buccal apparatus designed for processing large prey.

If one accepts *Pohlsepia*[34] as the earliest stem group octopod, this group would have originated in the Carboniferous. Doubt is cast on this hypothesis[14] by *Pohlsepia*'s poor preservation, the ten arms in *Jeletzkya douglassae* (coeval with *Pohlsepia*), and the absence of octopod fossils from at least the Permian and Triassic. According to molecular clock analyses by Tanner et al.[20], they did not originate before the Triassic, while Kröger et al.[1] proposed an Early Jurassic origin. Recently, the first evidence of early stem octopods was presented from the late Kimmeridgian[49,50].

From the specimens discussed herein and published material[28] seen in lateral aspect, it is evident that the aperture had a dorsal projection and a ventral (hyponomic) sinus. This is relevant because the broad ventral sinus provided some space for the development of the muscular mantle, for lateral movements of the hyponome and improved the maneuverability of *Gordoniconus* compared to orthocones with straight apertures. Also, the inclined aperture suggests that at least temporarily, the animal was able to tilt its body into an oblique or even horizontal position facilitated by the mass of the mineralized rostrum (perhaps supported by apically positioned fins or mantle folds).

More derived coleoids such as belemnitids and phragmoteuthids have proportionally shorter phragmocones with higher apical angles. Additionally, the thickness of mineralized shell in the rostrum increased in some lineages, thereby shifting the centers of mass and buoyancy closer towards each other[38]. This improved maneuverability in space and could have facilitated horizontal swimming, which is important for hunting and reproduction.

Coleoid ink is commonly fossilized[51,52]. We suggest that its absence in unequivocal specimens of *Gordoniconus* shows that the ink sac evolved after endocochlisation[29], as a defense organ, possibly due to the conch losing its protective function. Fossil ink sacs from later in the Carboniferous prove, however, that ink sacs evolved soon thereafter[29].

Fins are very rarely preserved throughout the Mesozoic coleoid fossil record[14,37,40,44,45]. Symmetrically arranged lateral furrows in rostra provide indirect evidence for fins in Mesozoic coleoids[53], while evidence from the Paleozoic is missing. Fins attached to the rostrum make sense as an adaptation for horizontal swimming and thus, we think that the fossil record of thick rostra in the Mesozoic reflects the presence of fins. It is conceivable that there were mantle wrinkles in Paleozoic forms that evolved into fins in Mesozoic coleoid lineages (and some Late Paleozoic ancestors).

How was the cephalopod conch internalized? There is a consensus that the Coleoidea derived from the Bactritida either in the Early Devonian[20,54] or in the Early Carboniferous[1]. This is relevant because the Bactritida were supposedly ectocochleate while the endocochleate state is one of the most important autapomorphies (apart from the ink sac) of coleoids. Consequently, mid-Paleozoic cephalopods must have evolved a shell-secreting tissue reaching to the apical end of the phragmocone (the initial chamber) or possessed it from early development and retained it until they reached maturity (Fig. 7). Evidence from Paleozoic and Mesozoic coleoids shows that their embryos had already deposited shell on the posterior of the initial chamber forming the subconical primordial rostrum[28,55,56]. Similar structures

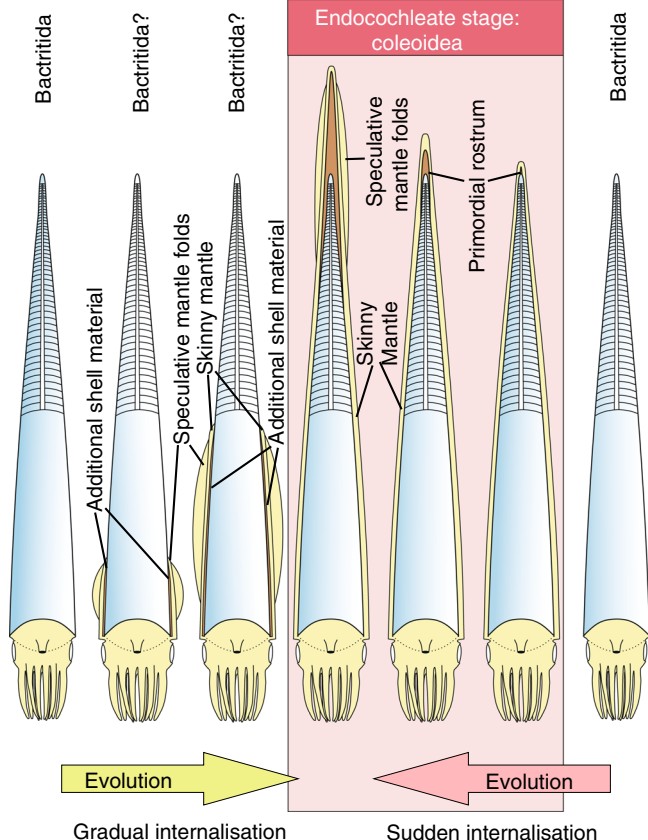

**Fig. 7** Two alternative explanations of endocochlisation. Owing to the patchy fossil record of early coleoids, it is unclear how the evolutionary transition from cephalopods with external (ectocochleate) to such with internal conchs (endocochleate) occurred. Here, we propose two models. Light blue—conch; light yellow—soft parts; brown—mineralized rostrum

documenting the early evolution of an internal conch have been documented from other Paleozoic fossils[9–13,28,34,51,57–61].

The strangeness of the evolution of an endocochleate conch becomes clear when taking the very long and slender conch of *Gordoniconus* into account, trying to imagine that living mantle tissue reached the tip of the rostrum all the way from the aperture. It is simple to understand how the embryonic conch became internal[56], but it is less obvious what could have been the selective advantage of conch internalization. Also, the evolutionary pathway leading to a conch that is endocochleate throughout ontogeny is unknown.

An internal conch has the disadvantage of exposing soft-tissues directly to predators and parasites, requiring secondary defense mechanisms (ink, flexible camouflage or poison, all of which evolved later in coleoids). Potential drivers of the early evolution of coleoids could have been the inability to repair shell damage from the outside, sensory functions and/or alteration of the distribution of mass as previously discussed and a near-apical attachment of fins (for hydrodynamic lift[62]), thereby making a nearly horizontal swimming position possible. This would have improved horizontal swimming abilities, which are important to catch prey, to escape predators, and to search for and reach mating partners. These factors represent powerful agents to counteract selective pressure, fostering evolutionary change in actively swimming marine animals.

Unfortunately, anatomical detail of the apical mantle is missing and the imprints behind the apical end of the holotype of *Gordoniconus* are likely landing marks rather than fins. Independent of the interpretation of these marks, fins would already have been

useful for hatchlings of early coleoids. For example, minute holoplanktic gastropods also possess fins, and there are some Thecosomata[63] possessing straight conchs. Although the evidence for evolutionary pathways is still poor, we suggest the following alternative hypotheses (Fig. 7):

(1) During middle or late ontogeny, the mantle of early stem-coleoids began to grow out of the aperture backwards eventually covering the entire phragmocone (possibly subsequently evolving keel-like folds for hydrodynamics), for shell repair from the outside and/or as a sensory surface (Fig. 8, top). Through evolution, the external mantle grew further apically and began to form increasingly earlier in ontogeny, ultimately reaching the embryonic stage. The advantage driving the positive selection likely applied to the hatchling as well, fostering ever-earlier appearance of conch internalization in ontogeny. Further heterochronic growth traits in coleoid cephalopods have been described[15,50,64].

The mantle contains muscles, as, e.g., in cypraeid gastropods (cowries). Their mantle can move to some extent, producing wrinkles and thinner or thicker parts. Speculatively, early coleoids probably also had similar mantle wrinkles that provided improvements in locomotion, particularly for steering and by producing a slight hydrodynamic lift. Feedbacks from positive selection for more strongly developed folds might have led to the evolution of increasingly muscular external mantle and fins. In turn, with the accumulation of shell-forming tissues, more carbonate was secreted at the rostrum, later leading to thick belemnite-like rostra. To what extent the thick mineralized rostrum helped the coleoid to achieve a horizontal position requires further research[31,38,62].

Proof for this course of character evolution would be a transitional form where a conch with bactritid morphology shows evidence of secondary formation of shell from the outside only around the body chamber with no secondary shell around the initial chamber. The bulk of Carboniferous taxa with external

(secondary) shell layers is missing clear evidence of a fully invested initial shell (e.g., *Palaeoconus*, *Shimanskya*, *Donovaniconus*); among Carboniferous coleoids, only *Hematites* (Late Mississippian) and *Mutveiconites* (Late Pennsylvanian) are confirmed to be internal starting in early embryonic stages[28,65]. The enveloped initial shell of *Hematites* shows that the coleoid conch internalization was completed by the end of the Late Mississippian (310 my). Forms intermediate between partially endocochleate bactritoids and fully endocochleate coleoids may therefore be expected in early Mississippian sediments.

(2) A mutation caused the shell-secreting tissue to surround the embryonic conch from a very early embryonic stage (Fig. 8, bottom). A comparable transition was reported for gastropods[66]. This would have represented some kind of pre-adaptation, where either the endocochleate condition simply lasted or the endocochleate state extended into increasingly later ontogenetic stages through phylogeny.

Proof for this hypothesis would be a transitional form showing secondary shell deposited from the outside during the embryonic phase. Such findings have been published by Doguzhaeva et al.[10–12,61], suggesting that endocochlisation may have evolved convergently several times in different cephalopod lineages (ammonoids and nautiloids[67–69]). Thus, we favor the hypothesis that the conch became endocochleate already in an early embryonic stage.

What are the implications for the paleoecology of other Paleozoic cephalopods? The superficial similarities in conch morphology between *Gordoniconus* and bactritids suggest some similarities in their modes of life. However, *Gordoniconus* had a rostrum, which is absent in bactritids. For both groups, an oblique to nearly vertical position in the water column appears probable when inactive. The presence of conodont and fish remains in its digestive tract[9] supports the hypothesis that, being closely related, bactritids (and ammonoids) were micro-predators as well, feeding on small animals and perhaps also carrion. In turn, this suggests that an approximately horizontal orientation could be achieved at least occasionally by *Gordoniconus* and other early rostrum-bearing coleoids during life[38,62]. Both prey groups are known to have been good swimmers and while one of these prey animals could have been carrion, it is unlikely that both were. Notably, both larger prey (fish) and smaller prey (conodonts) were ingested. That both were broken into small bits can be explained by the biting power of the buccal apparatus and limitations in size present on the oral opening such as the necessity that these items had to pass through the cephalic cartilage surrounding the esophagus.

The prod mark apical to the rostral tip of *Gordoniconus* implies that the rostrum accumulated enough mass to let the carcass sink apex first into the sediment. This supports our contention that *Gordoniconus* lived in the water column. Similar observations are widely documented from Jurassic belemnites[48], which occasionally were embedded obliquely to vertically with the apex pointing downward. This would suggest indirectly that an approximately horizontal orientation could be achieved by the animal during life[38,62].

In conclusion, we examined the holotype, a paratype and some topotypes of *Gordoniconus beargulchensis*, the earliest coleoid preserved with its complete conch, jaws, and soft parts. Some of these body parts have been recognized earlier[9,28], but we were able to refine those interpretations by examining the fossils under light of various wavelengths and geometry, including UV-light and white light applied at a very shallow angle (including RTI), as well as X-ray elemental maps.

These new images revealed the presence of probably ten undifferentiated and short arms. A supposedly hook-bearing tentacle[28] is re-interpreted as a coprolite. Additionally, we recognized remains of the cephalic cartilage, further details of the

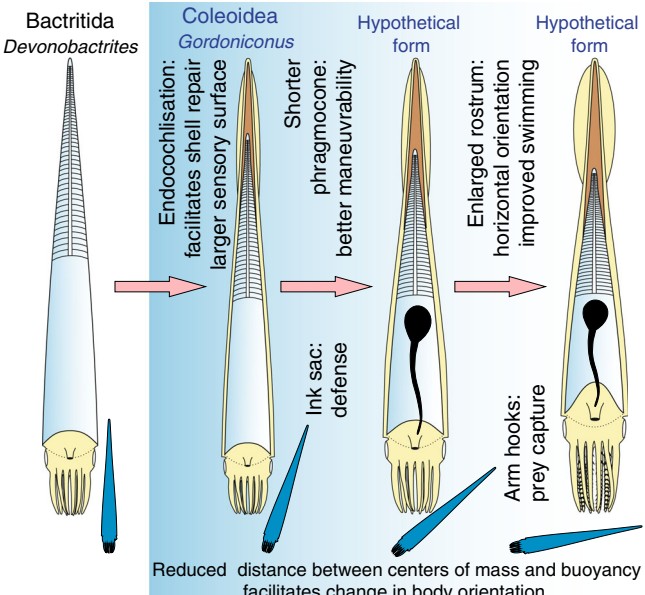

**Fig. 8** Hypothetical temporal sequence of key-innovations that occurred in the evolution of coleoids within the Carboniferous. Note that a fossil record of fins or similar structures is missing from Paleozoic coleoids; for all other depicted body parts, fossil evidence is available from the Carboniferous. The small blue cephalopods shall represent the syn vivo orientation of these animals, changing from vertical to approximately horizontal. Light blue—conch; light yellow—soft parts; dark gray—ink sac; brown—mineralized rostrum

digestive tract and possibly the gills (or gonads). Approximately symmetric imprints posterior to the apex of the rostrum represent a landing mark that formed when the carcass of the animal sank onto the sediment surface, as documented from Mesozoic belemnites.

The presence of a distinct conical rostrum indicates the endococheate state of the conch and corroborates the interpretation of this species as a coleoid. Based on this fact, we introduce the hypothesis that the internalization of the conch began to support horizontal swimming movements very early in coleoid evolution by shifting the center of mass closer to the center of buoyancy to increase maneuverability. We speculate that the development of an external mantle may have included the formation of longitudinal wrinkles or ridges, which acted as a basis for a primordial fin that then became vital for crown coleoids as their body position became more horizontal. Evolutionary feedbacks between locomotory advantages and anatomy positively selected for thicker rostra, ventral reduction of the body chamber wall, differentiated mantle folds, and eventually true fins (at the latest in the Jurassic[37,38,40,47]).

We propose that the internalization of the conch was the main first evolutionary step in the origination of coleoids that happened at the latest in the Early Carboniferous. Innovations such as ink sac (Early Carboniferous) and fins (Fig. 8) followed later and tentacles represent comparatively modern modifications of the arm crown (Jurassic).

## Methods

**Light of varying wavelength**. In order to reveal previously undetected anatomical details, we used various light sources. Among these was a regular UV-lamp. The camera lens was equipped with a polarization filter in order to remove the reflected UV-light. We further treated these images using PhotoShop (we automatically corrected the color and increased the contrast) in order to reduce the color artifact and to enhance the contrast. Additionally, we used a regular white lamp, which was installed at a minimal angle to enhance shadows of very small elevations on the fossil. Other details were revealed by combining different reflectance and luminescence images collected within narrow regions of the visible and near-infrared spectrum (defined by narrow band-pass filters) using different excitation wavelengths from the UV-A (365–400 nm) to the near-IR (~800 nm).

**Synchrotron micro-X-ray fluorescence**. µXRF mapping was performed at the DiffAbs beamline of the SOLEIL synchrotron source (France) using a monochromatic beam of 18.2 keV, selected for excitation of K-lines from phosphorus to zirconium and L-lines from cadmium to uranium. The beam was focused down to 10 µm using Kirkpatrick-Baez (KB) mirrors. The sample was mounted on a scanner stage allowing few centimeters movements with micrometer accuracy. Incident X-ray beam impinges on the sample at an angle of 65° from the surface and the fluorescence radiation is detected with a mono element silicon drift detector (Vortex EX, total active area: 100 mm²) at an angle of 25° from the surface. The anterior part of AMNH 50267 was mapped at a 80 µm lateral resolution, with a 50 ms dwell time. µXRF elemental maps were then produced through the collection of integrated intensities in selected spectral regions of interest corresponding to Fe, Zn, Sr, and Y Kα emissions.

**Reflectance transformation imaging**. RTI[70,71] was applied to the holotype of *Gordoniconus beargulchensis* using the RTI-dome in the Digitization Lab of the Institute for the Preservation of Cultural Heritage at Yale University, New Haven, Connecticut. The software for processing is RTIBuilder and the software for viewing is RTIViewer. The RTI-dome is dome-shaped and carries 100 regularly arranged lamps. Photos are taken from the azimuth position with each light switched on once, resulting in 100 photos combined in one large file. The obtained files can be used to change lighting while viewing the images by moving the cursor. This is relevant in the material described here because only very low light from various directions shows some of the described structures.

**Statistics and reproducibility**. No statistical analyses were performed. Data were based on one main specimen and three additional specimens depicted in the supplement. All photographs can be reproduced by using the same RTI-settings or, respectively, light sources of the same wavelengths. The RTI-file is composed of 100 photos taken with a standardized setting. Accordingly, repetition of photography will yield nearly identical results.

**Reporting summary**. Further information on research design is available in the Nature Research Reporting Summary linked to this article.

## Data availability

All data generated or analyzed during this study are included in this published article (and its supplementary information files). The specimen is kept in the American Museum of Natural History in New York, with the numbers AMNH 43264 (part) and AMNH 50267 (counterpart). Reflectance Transformation Images, *.rti and *.ptm of the holotype AMNH 43264 are provided as Supplementary Data. To view these images and change the illumination, the required software RTIViewer is available at http://culturalheritageimaging.org/What_We_Offer/Downloads/View/. These RTI and PTM files have been deposited in Figshare[72,73].

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

## Acknowledgements

This study is a contribution to project Nr. 200021_169627 funded by the Swiss National Science Foundation. Rosemarie Roth (Zürich) kindly helped with photography of the holotype. We acknowledge the SOLEIL Synchrotron for provision of synchrotron radiation facilities and we would like to thank the DiffAbs beamline team and Clément Jauvion (MNHN, Paris) for assistance during the experiments. We greatly appreciate the technical help provided by the photographers Mariah Slovacek and Steve Thurston (American Museum of Natural History, New York). We thank Jessica Utrop (Yale University, New Haven, Connecticut) who examined the specimen using RTI and Carmella Cuomo (University of New Haven, Connecticut) who helped in the interpretation of the results. Kallie Moore (University of Montana Paleontology Center Research Collection Missoula, Montana) and Albert Kollar (Carnegie Museum of Natural History, Pittsburgh, Pennsylvania) kindly put specimens at our disposal. We are grateful to W. Bruce Saunders (Bryn Mawr, Pennsylvania) for the additional information he provided on Bear Gulch material. We appreciate the thorough reviews by Kenneth De Baets (Erlangen) and Thomas Clements (Cork).

## Author contributions

C.K. conceived the paper, took some of the photos, and wrote much of the text. N.L. organized materials, provided the RTI-data. N.L., D.F., R.M., A.P. and R.H. contributed their knowledge on different groups of cephalopods and wrote parts of the text. P.G. and

S.R. carried out the synchrotron work and made some of the images. All authors contributed to and revised the manuscript.

## Additional information

**Competing interests:** The authors declare no competing interests.

