## [Peer Review File · Communications Biology]

Reviewers' comments:

Reviewer #1 (Remarks to the Author):

The paper is well-written, applies innovative methods and presents new evidence for the anatomy of early coleoids. This is interesting for evolutionary (paleo)biologist, ecologist (as they present stomach contents) and biologist/taxonomist working on shell internalization. I only found some minor aspects which could further improve the manuscript and make it an even more important contribution to these fields:

p. 5, : Interpretation of arm crown: it would be worth stressing that in this kind of preservation it is likely that the arms (or at least arm stumps) would be preserved based on decay experiments (Clements et al. 2017). This would be worth briefly stating as it further supports you're your interpretation is likely.

p. 6: when arguing for ten short arms it would be worth stressing that the preserved are similar in size which further supports your idea.

p. 11: "we think that it is a landing mark", better would be "it is more consistent with a landing mark"

p. 11: "An important finding ... ten similar arms." I understand what you are coming from, but technically this is not correct, you only find evidence for six preserved arms not ten. The preservation of your specimens support ten arms but they are not preserved/visible! I agree with your interpretation so I suggest rephrasing this. This could be done in various ways. You could say there are at least six similar arms which could only cover 4 additional similar sized arms or you could say that the fossil preservation is consistent with the ancestral condition of ten similarly sized arms.

p. 11: Pohlsepia. Various authors doubt the assignment of Pohlsepia and it might be appropriate citing some in this context.

p. 11: Ink sacs from the Carboniferous – this is confusing, do mean later in the Carboniferous and/or in more derived forms? This should be stated.

P. 15: "(2) A mutation" It this context (and maybe earlier in the introduction), it would be worth citing the following references which highlights that snails can be turned into slugs by intervening in early ontogeny:

Osterauer, R., Marschner, L., Betz, O., Gerberding, M., Sawasdee, B., Cloetens, P., ... & Köhler, H. R. (2010). Turning snails into slugs: induced body plan changes and formation of an internal shell. *Evolution & development*, 12(5), 474-483.

The following quote from the introduction of Osterauer et al. (2010) would also be consistent with your hypothesis: "Even though mechanisms of shell internalization vary between different evolutionary lines, always the interactions between mantle and shell growth have been shown to be modified in the early individual ontogeny."

These comments can also be found in the annotated pdf.

Kenneth De Baets

Reviewer #2 (Remarks to the Author):

I really enjoyed reading this paper, it is well structured and reads very well and the figures are marvellous. I highly recommend this paper for publication, it tackles a very important topic regarding early coleoid evolution, and goes a significant way to highlighting an important re-evaluation of a fossil that shows transitional evolutionary morphological characters. I am convinced by the author's novel interpretations and their diligent care not to 'over interpret' this fossil. I believe this paper will significantly add to the current debate on coleoid evolution.

Most of my edits are purely grammatical (I hope you don't mind that I offered some suggestions), but there are a few little things I would like to recommend that will make the manuscript more complete:

1) I think the introduction could do more to explain the current hypotheses of coleoid evolution in a review format. This would make the paper have wider scope for a generalist audience and I feel it would attract more citations in the long run. This could be done in the text or in a figure? I don't know if it is outside the scope of the authors' work, but would it be possible to place *Gordoniconus* into a tree to help the readers here?

2) At no point in the text (even in the methods) is RTI explained, except in a supplementary figure caption... I had to look it up. It needs to be described and in the first use the full name needs to be used,

3) Figure 5 is a lovely drawn figure but the layout is very chaotic. I have added suggestions to make it clearer.

I have no comments on the methods. The authors' seem to have been careful and very thorough. I don't think the synchrotron data adds much and maybe SEM EDX or XRD would have yielded better results, but I imagine this fossil is rather large for most environment-chambers. I personally do not believe these analyses would add much to the paper and do not recommend them, they just would be a nice compliment.

I must state again how much I enjoyed reading this paper. I look forward to citing in the future!
Thomas Clements.

(If any of my comments are unclear, I am happy to be contacted by the authors for clarification).

Reviewer #3 (Remarks to the Author):

An exceptionally thorough and interesting paper. I believe that all possible methods were carried out and the resulting images, coupled with detailed analysis and well justified arguments produce an important step forward in our understanding. My only query is that of the argument of the gonad/ gills. While it is exceptionally well justified as to why the authors have gone with the structure being gills, the location in the animal do unsettle me slightly from a functional perspective. However the argument that they look structurally like gills is well reasoned. Perhaps this will not be fully resolved until (if?) we find more material with better/ differing preservation showing unambiguous gills.

Comments on the Reviewers' comments:

Reviewer #1 (Remarks to the Author):

p. 5, : Interpretation of arm crown: it would be worth stressing that in this kind of preservation it is likely that the arms (or at least arm stumps) would be preserved based on decay experiments (Clements et al. 2017). This would be worth briefly stating as it further supports you're your interpretation is likely.

Reference and remark was added.

p. 6: when arguing for ten short arms it would be worth stressing that the preserved are similar in size which further supports your idea.

A remark was added accordingly.

p. 11: "we think that it is a landing mark", better would be "it is more consistent with a landing mark"

Done.

p. 11: "An important finding ... ten similar arms." I understand what you are coming from, but technically this is not correct, you only find evidence for six preserved arms not ten. The preservation of your specimens support ten arms but they are not preserved/visible! I agree with your interpretation so I suggest rephrasing this. This could be done in various ways. You could say there are at least six similar arms which could only cover 4 additional similar sized arms or you could say that the fossil preservation is consistent with the ancestral condition of ten similarly sized arms.

We have more carefully phrased this part to express the uncertainty.

p. 11: Pohlsepia. Various authors doubt the assignment of Pohlsepia and it might be appropriate citing some in this context.

We have changed the text to indicate the doubtfulness of this species.

p. 11: Ink sacs from the Carboniferous – this is confusing, do mean later in the Carboniferous and/or in more derived forms? This should be stated.

Done.

P. 15: "(2) A mutation" In this context (and maybe earlier in the introduction), it would be worth citing the following references which highlights that snails can be turned into slugs by intervening in early ontogeny:

Osterauer, R., Marschner, L., Betz, O., Gerberding, M., Sawasdee, B., Cloetens, P., ... & Köhler, H. R. (2010). Turning snails into slugs: induced body plan changes and formation of an internal shell. *Evolution & development*, 12(5), 474-483.

The reference was included.

Reviewer #2 (Remarks to the Author):

Most of my edits are purely grammatical (I hope you don't mind that I offered some suggestions),

We followed the suggestions of this reviewer of the annotated pdf.

1) I think the introduction could do more to explain the current hypotheses of coleoid evolution in a review format. This would make the paper have wider scope for a generalist audience and I feel it would attract more citations in the long run. This could be done in the text or in a figure? I don't know if it is outside the scope of the authors' work, but would it be possible to place *Gordoniconus* into a tree to help the readers here?

We have added a short paragraph giving an overview over the main evolutionary events.

2) At no point in the text (even in the methods) is RTI explained, except in a supplementary figure caption... I had to look it up. It needs to be described and in the first use the full name needs to be used,

We have added explanations in the abstract and in the methods section.

3) Figure 5 is a lovely drawn figure but the layout is very chaotic. I have added suggestions to make it clearer.

We did not change the arrangement of the single drawings BUT we added arrows to clarify the logical context, i.e. photo to camera lucida-drawing to combination of both drawings to interpretative sketch.

Reviewer #3 (Remarks to the Author):

My only query is that of the argument of the gonad/ gills. While it is exceptionally well justified as to why the authors have gone with the structure being gills, the location in the animal do unsettle me slightly from a functional perspective. However the argument that they look structurally like gills is well reasoned. Perhaps this will not be fully resolved until (if?) we find more material with better/ differing preservation showing unambiguous gills.

Well, I guess the way we discussed this is honest enough and the reader can make up his/ her own mind.